# Non-Hermitian quantum quenches in holography

Sergio Morales-Tejera[1,2*] and Karl Landsteiner[1†]

**1** Instituto de Física Teórica UAM/CSIC, c/Nicolás Cabrera 13-15,
Campus de Cantoblanco, 28049 Madrid, Spain
**2** Departamento de Física Teórica, Universidad Autónoma de Madrid,
Cantoblanco, 28049 Madrid, Spain

* sergio.moralest@uam.es, † karl.landsteiner@csic.es,

## Abstract

The notion of non-Hermitian $\mathcal{PT}$ symmetric quantum theory has recently been generalized to the gauge/gravity duality. We study the evolution of such non-Hermitian holographic field theories when the couplings are varied with time with particular emphasis on the question non-unitary time vs. unitary time evolution. We show that a non-unitary time evolution in the dual quantum theory corresponds to a violation of the Null Energy Condition (NEC) in the bulk of the asymptotically AdS spacetime. We find that upon varying the non-Hermitian coupling the horizon of a bulk AdS black hole shrinks. On the other hand varying the Hermitian coupling in the presence of a constant non-Hermitian coupling still violates the NEC but results in a growing horizon. We also show that by introducing a non-Hermitian gauge field the time evolution can be made unitary, e.g. the NEC in the bulk is obeyed and an exactly equivalent purely Hermitian description can be given.

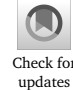

# 1   Introduction

One of the core axioms of quantum mechanics is that observable quantities are represented by Hermitian operators. It is then a bit surprising that some seemingly non-Hermitian Hamiltonians can have real energy spectra. In [1–4] it has been realized that the underlying reason for this is that such Hamiltonians have an anti-linear $Z_2$ symmetry commonly denoted by $\mathcal{PT}$. In simple cases it can be thought of as the product of parity and time-reversal. $\mathcal{PT}$ symmetric quantum mechanics has raised much interest and is reviewed in the recent book [5]. The basic properties of a $\mathcal{PT}$ symmetric Hamiltonian are that there are two regimes separated by a $\mathcal{PT}$ critical point. In the so-called $\mathcal{PT}$ symmetric regime the eigenvectors of the Hamiltonian are simultaneous eigenvectors of $\mathcal{PT}$ which implies that the energies are real. In the $\mathcal{PT}$ broken regime the eigenvectors of the Hamiltonian come in doublets under the $\mathcal{PT}$ symmetry with energy eigenvalues that are complex conjugate to each other.

It has also been pointed out that in the $\mathcal{PT}$ symmetric regime the Hamiltonian is pseudo-Hermitian, i.e. there exists a Hermitian similarity transformation such that $\eta H \eta^{-1} = h$ with $h^\dagger = h$ being Hermitian [6–9]. An early example of this was investigated by Dyson in [10] and thus $\eta$ is sometimes called the "Dyson map". This observation allows to construct a $\mathcal{PT}$ symmetric Hamiltonian starting from a Hermitian one. A simple example for a two dimensional Hilbert space is a follows. Every Hermitian Hamiltonian acting on a two dimensional Hilbert space can be written as a linear combination of Pauli matrices

$$h = \sum_{i=1}^{3} g_i \sigma_i \,. \tag{1.1}$$

A physically equivalent Hamiltonian can be obtained by a unitary transformation with $U = \exp(i\alpha\hat{n}.\vec{\sigma}/2)$ which is just a rotation by an angle $\alpha$ around the axis defined by a unit vector $\hat{n}$. It is also possible to generate an equivalent non-Hermitian Hamiltonian by analytically continuing the angle $\alpha = i\beta$. This gives the Dyson matrix $\eta = \exp(\beta\hat{n}.\sigma/2)$ and defines a non-Hermitian Hamiltonian. As concrete example consider

$$H = g\sigma_x + i\Gamma\sigma_z \,, \tag{1.2}$$

with eigenvalues $E_\pm = \pm\sqrt{g^2 - \Gamma^2}$ and $\mathcal{PT}$ symmetry $\sigma_x K$, where $K$ is complex conjugation. Obviously the energy eigenvalues are real as long as $|\Gamma| < |g|$. Setting $g = g_0\cosh(\beta)$ and $\Gamma = g_0\sinh(\beta)$ it is clear that the non-Hermitian Hamiltonian is generated from the Hermitian $h = g_0\sigma_x$ by a similarity transformation $\eta = \exp(\beta\sigma_y/2)$. The critical point $g = \Gamma$ can be reached by taking $\beta \to \infty$ and simultaneously $g_0 \to 0$ keeping the product fixed. On the other hand the $\mathcal{PT}$ - broken regime with $|g| < |\Gamma|$ can not be reached from the Hermitian Hamiltonian by any similarity transformation or limits thereof.

We now make a key observation which allows to translate the concept of $\mathcal{PT}$ symmetry to the gauge/gravity duality in a straightforward way. Since the Dyson map is an analytically continued rotation around the $y$-axes we can write the final Hamiltonian as $H = g_i R_{ij}\sigma_i$ with the rotation denoted by $R_{ij}$. This makes it obvious that we can also define the new Hamiltonian by a transformation acting on the couplings $g_i$ rather than acting directly on the Hilbert space.

Indeed starting from the vector of couplings $\vec{g}_0 = (g_0, 0, 0)$ the new non-Hermitian couplings are obtained by a hyperbolic rotation by the hyperbolic angle $\beta$.

While the $\mathcal{PT}$ symmetric regime is therefore mathematically equivalent to quantum mechanics with a Hermitian Hamiltonian the physical interpretation is quite different. A standard Hermitian Hamiltonian describes a closed isolated quantum system whereas a $\mathcal{PT}$ symmetric Hamiltonian describes an open quantum system with an exact balance between inflow and outflow. The Hamiltonian (1.2) can be understood as the one for a particle hopping between two locations with amplitude $g$ and where in the first location the particle can escape to an open environment at a rate of $2\Gamma$ whereas in the other location identical particles can enter the system at precisely the same rate. As long as $|\Gamma| < |g|$ the system is in a steady state since a surplus particle which just has entered, will hop to the other location and leave the system before another particle can enter. On the other hand if $|\Gamma| > |g|$ particles escape faster than new particles can enter and hop to the leaky location. So the first location will be emptied whereas particles will accumulate without bound in the other location.

Let us suppose now that an experimenter can manipulate the leak/growth rate $\Gamma$ and the hopping rate $g$ in a time dependent manner while keeping always the exact balance between entrance and escape rates. In this situation the non-Hermitian Hamiltonian at any time $t$ would be quasi-Hermitian

$$H(t) = \eta(t)^{-1} h(t) \eta(t), \tag{1.3}$$

with $\eta$ determined by a time dependent, hyperbolic angle $\beta(t)$. Is such an open and time dependent quantum system still equivalent to a usual Hermitian time dependent quantum system? There are basically two radically different answers to this question.

We can first simply consider the time dependent Schrödinger equation based on the Hamiltonian (1.3)

$$i\partial_t |\Psi\rangle = H |\Psi\rangle . \tag{1.4}$$

If we think of the experimenter manipulating the hopping parameter $g$ and the leak/growth rate $\Gamma$ in a time dependent way this seems indeed the natural guess of describing how the open quantum system evolves in time. This is however problematic since the time evolution does not respect unitarity. To see this we note that at any given moment the Hilbert space can be mapped to the one of the Hermitian Hamiltonian $h(t)$ by $|\Psi\rangle = \eta^{-1} |\psi\rangle$. Furthermore it is easy to see that the following product is equivalent to the standard Hermitian product of vectors

$$\langle \Psi | \Phi \rangle_\eta := (\langle \Psi | \eta^2) |\Phi\rangle = \langle \psi | \varphi \rangle . \tag{1.5}$$

In the time dependent case this product is not conserved under the time evolution (1.4)

$$\partial_t(\langle \Psi | \Phi \rangle_\eta) = 2\langle \Psi | (\eta^{-1} \partial_t \eta) \Phi \rangle_\eta , \tag{1.6}$$

where one uses $H^\dagger \eta^2 = \eta^2 H$. So one necessarily is faced with a non-unitary time evolution!

A second point of view leaves at least a formal cure to this problem. If we define the modified time dependent Schrödinger equation

$$i(\partial_t + \eta^{-1} \dot{\eta}) |\Psi\rangle = H |\Psi\rangle , \tag{1.7}$$

then indeed we have with this new rule for updating state vectors

$$\partial_t(\langle \Psi | \Phi \rangle_\eta) = 0 . \tag{1.8}$$

A mathematically natural interpretation is that we have introduced a time component of a gauge field for the gauged Schrödinger equation

$$iD_t |\Psi\rangle = i(\partial_t - iA_t) |\Psi\rangle = H |\Psi\rangle . \tag{1.9}$$

Treating the time dependent similarity transformation therefore as a gauge transformation one sees that the Hermitian system described by $h$, $|\psi\rangle$, $A_t = 0$ is gauge equivalent to $H = \eta^{-1}h\eta$, $|\Psi\rangle = \eta^{-1}|\psi\rangle$ and $A_t = i\eta^{-1}\partial_t\eta$. In particular for our two dimensional toy model $A_t = i\partial_t\beta\sigma_y/2$.

It is a priori impossible to decide which of the two possibilities of time evolution (1.4) or (1.7) is the unique "correct" one. The second one is mathematically equivalent to a standard unitary time evolution based on a Hermitian Hamiltonian, but it is not necessarily the unique physically correct one. We stress that even in the time independent case the non-Hermitian Hamiltonian describes a physically different system: open with balanced gain/loss whereas the Hermitian Hamiltonian describes a closed quantum system. Therefore we advocate the point of view that an experimenter who manipulates the gain/loss rate also would need to do some additional manipulation corresponding to switching on the gauge field $A_t = i\eta^{-1}\partial_t\eta$ in order to maintain a unitary time evolution. In our case the form of the gauge field $A_t = i\partial_t\beta\sigma_y/2$ suggests that the forward hopping amplitude has to be different form the backward hopping amplitude. In this way non-Hermitian hopping amplitudes are introduced which compensate for the change in the leak/gain rates in precisely the correct way as to maintain a steady state equilibrium. This in turn allows the mapping the Hamiltonian back to a Hermitian time dependent one. In this way the non-Hermitian "gauge" field has a physical interpretation. While formally it appears like a pure gauge it has a non-trivial physical consequence. This is of course in stark contrast to real gauge fields in which a pure gauge has no physical consequence. If, on the other hand, the experimenter only manipulates the Hermitian hopping amplitude and/or the gain/loss rates then the time evolution is described by (1.4). It will be necessarily non-unitary and thus reflecting the openness of the quantum system. Time dependent $\mathcal{PT}$ quantum mechanics has been reviewed recently in [11] and [12].

After these introductory remarks on the issues involved in time dependent $\mathcal{PT}$ symmetric quantum mechanics we now want to briefly outline the motivation for the present research. There are a number of attempts to generalize the concept of $\mathcal{PT}$ symmetry to quantum field theories, some recent works include [13–21]. In particular [22] investigates the consequences of a space-dependent non-Hermitian gauge field. For a recent review of model building efforts with non-Hermitian quantum field theories see also [23]. Quantum field theory provides of course additional technical and conceptual difficulties when compared to simple quantum mechanical systems. In particular time dependent Hamiltonians (or Lagrangians) present already quite some technical difficulties in the usual Hermitian setup and call for advanced tools of non-equilibrium quantum field theory [24, 25].

Surprisingly it is considerably less difficult to deal with some strongly coupled quantum field theories which allow a holographic description in terms of a (classical) gravitational theory in one higher dimension. For good introductions to the gauge/gravity duality see [26, 27]. The application of the gauge/gravity duality to out-of-equilibrium physics has lead to an improved understanding of relativistic hydrodynamics and might even be relevant to real world experimental observations since it allows to model the time evolution of the quark gluon plasma in heavy ion collisions [21, 28–31].

What makes the gauge/gravity duality so powerful in the investigation of non-equilibrium phenomena is that one can easily implement time dependence of the couplings of the dual strongly coupled field theory by imposing time dependent boundary conditions on the fields in an asymptotically Anti de-Sitter spacetime. This motivates us to investigate the question of time evolution in $\mathcal{PT}$ symmetric quantum field theory in a strongly coupled setup within the gauge/gravity paradigm. A holographic model with $\mathcal{PT}$ symmetry has been developed in [32]. There it was shown that indeed in holography the two regimes, $\mathcal{PT}$ symmetric and $\mathcal{PT}$ broken, are realized and separated by a critical point. The key ingredient was to generalize the boundary conditions on the fields at asymptotic infinity to non-Hermitian ones by

a complexified $U(1)$ transformation. Here we would like to emphasize that non-Hermitian boundary conditions on fields are quite common in the gauge/gravity duality. In particular in the study of quasinormal modes one imposes infalling boundary conditions at the horizon of an asymptotically Anti de-Sitter black hole. These infalling boundary conditions break Hermiticity and are responsible for the eigenfrequncies to be complex numbers. From this point of view a holographic $\mathcal{PT}$ theory is different in the fact that the Hermiticity breaking boundary conditions are imposed not at the horizon but at asymptotic infinity. The physical interpretation is also different. The infalling boundary conditions on the horizon lead to dissipation and diffusion typical to finite temperature field theory. On the other hand the non-Hermitian but $\mathcal{PT}$ symmetric boundary conditions at the asymptotic boundary of Anti de-Sitter space signal that the quantum system under consideration is open and suffers from a balanced in- and outflow to and from an environment.

The article is organized as follows. In section two we introduce the holographic $\mathcal{PT}$ model and briefly review the results of [32]. In section three we present the results of the numerical simulations. There are various different situation to consider. First we study quenches in the purely non-Hermitian direction, i.e. the coupling to the non-Hermitian operator is time dependent. We show that generically the null energy condition is violated on the horizon. Then we study quenches which include excursions into the $\mathcal{PT}$ broken regime. It turns out that one can stay some finite amount of time in the $\mathcal{PT}$ broken regime and them go back into the $\mathcal{PT}$ symmetric regime and settle down at a static equilibrium solution. We also study periodic variations of the non-Hermitian coupling and find that the system evolves towards formation of a naked singularity. Then we study quenches in the purely Hermitian direction but with a constant non-vanishing non-Hermitian operator switched on. In this case the null energy conditions is still violated in the bulk but not on the horizon. Consequently the horizon grows similarly to a usual Hermitian quench. Finally we also introduce a non-Hermitian gauge field and show explicitly that the resulting time evolution is exactly equivalent to a conventional Hermitian quench. We summarize our conclusion in section four and present some technical details in the appendices.

## 2 Holographic model

A minimalistic approach to construct a holographic model for a non-Hermitian quantum field theory is to consider a Hermitian quantum field theory defined by its holographic dual and render both non-Hermitian by performing analogous manipulations in either side. This has been carried out in [32]. Following this approach we study the theory defined by the action

$$S = \frac{1}{2\kappa^2} \int_{\mathcal{M}} d^4x \sqrt{-g} \left[ R + \frac{6}{L^2} - \overline{D_\mu \phi} D^\mu \phi - m^2 \phi \overline{\phi} - \frac{V}{2} \phi^2 \overline{\phi}^2 - \frac{1}{4} F_{\mu\nu} F^{\mu\nu} \right] + S_{GHY} + S_{ct} , \tag{2.1}$$

where $S_{GHY}$ is the Gibbons-Hawking-York boundary term to make the variational problem well defined, $S_{ct}$ are the renormalization counterterms, $L$ is the AdS radius and $\kappa^2$ is the Newton constant. We work with the mostly plus metric. The field content consists of a $U(1)$ gauge field $A_\mu$, with field strength $F = dA$, a complex massive scalar field with charge $q$ under the gauge symmetry and the metric $g_{\mu\nu}$. That the fields are charged under the gauge symmetry is interpreted as having a symmetry in the quantum field theory under which the couplings also transform non trivially. We remind the reader that the couplings in the dual field theory are defined by the boundary conditions at asymptotic infinity. The Lagrangian (2.1) is thus left invariant but we land in a different theory, inasmuch as the coupling constants have changed. The covariant derivative $D_\mu$ is defined to act on the scalar field as $D_\mu = \partial_\mu - iqA_\mu$ and the overbar denotes complex conjugation. Finally, the quartic term in the potential for the scalar

field is required to find regular zero temperature domain wall solutions interpolating between two *AdS* geometries [32].

The equations of motion derived from the action read:

$$\frac{1}{\sqrt{-g}}\partial_\mu\left(\sqrt{-g}\,\overline{D^\mu\phi}\right)+iqA_\mu\overline{D^\mu\phi}-m^2\overline{\phi}-V\overline{\phi}|\phi|^2=0\,,$$

$$\frac{1}{\sqrt{-g}}\partial_\mu\left(\sqrt{-g}\,D^\mu\phi\right)-iqA_\mu D^\mu\phi-m^2\phi-V\phi|\phi|^2=0\,,$$

$$\frac{1}{\sqrt{-g}}\partial_\mu\left(\sqrt{-g}\,F^{\mu\nu}\right)-2q^2A^\nu|\phi|^2+iq\left(\phi\partial_\mu\overline{\phi}-\overline{\phi}\partial_\mu\phi\right)=0\,,\tag{2.2}$$

$$R_{\mu\nu}-\frac{1}{2}g_{\mu\nu}\left(R+\frac{6}{L^2}-\overline{D_\alpha\phi}D^\alpha\phi-m^2\phi\overline{\phi}-\frac{V}{2}\phi^2\overline{\phi}^2-\frac{1}{4}F_{\alpha\beta}F^{\alpha\beta}\right)$$
$$-\overline{D_{(\mu}\phi}D_{\nu)}\phi-\frac{1}{2}F_{\mu\alpha}F_\nu{}^\alpha=0\,,$$

with the indices in parenthesis denoting the symmetric part of the tensor. Note that the equations of motion are invariant under the $U(1)$ gauge transformation $A_\mu\to A_\mu+\partial_\mu\alpha(x)$, $\phi\to e^{iq\alpha(x)}\phi$ as they should. We set at this point the unphysical scale $L$ to 1. The mass parameter and charge of the scalar field are chosen to be $m^2=-2$ and $q=1$. This model is basically the same as the one used to study holographic superconductors in [33]. Our case differs in that we explicitly break the bulk gauge symmetry by sourcing the scalar field via non-trivial boundary condition.

We will solve the equations of motion with asymptotically anti-de Sitter (*AAdS*) boundary conditions on the metric. More concretely, we have chosen to write the metric ansatz in infalling Eddington-Finkelstein coordinates, with $v$ and $u$ denoting the temporal and radial coordinates respectively. The boundary is located at $u=0$, and the boundary coordinates are collectively denoted as $x$

$$ds^2=-f\,dv^2-\frac{2}{u^2}e^g du\,dv+S^2(dx_1^2+dx_2^2)\,,\tag{2.3}$$

where $(f,g,S)$ are functions of $(u,v)$ whose asymptotic (near boundary) expansion for small $u$ recover the *AdS* geometry

$$\lim_{u\to0}(u^2f)=1\,,\quad\lim_{u\to0}g=0\,,\quad\lim_{u\to0}S=1\,.\tag{2.4}$$

In order to render the problem non-Hermitian recall that the leading (non-normalizable) term $\phi_1(x)$ in the asymptotic expansion of each scalar field with our choice of mass is

$$\phi(u,x)=u\phi_1(x)+O(u^2)\,,\tag{2.5}$$

$$\bar{\phi}(u,x)=u\bar{\phi}_1(x)+O(u^2)\,.\tag{2.6}$$

The leading term takes the role of a coupling, i.e. $\phi_1$, $\bar{\phi}_1$ acts as the source for a charged scalar operator $\mathcal{O}$, $\bar{\mathcal{O}}$ of conformal dimension $\Delta=2$. Since our scalar field is complex we can write the leading term in the polar form $\phi_1(x)=e^{i\alpha(x)}\psi_1(x)$ with $\psi(x)$ real-valued. Accordingly, the leading term in the expansion of the complex conjugate field $\overline{\phi}(x,u)$ is simply $\overline{\phi_1}(x)=e^{-i\alpha(x)}\psi_1(x)$. At this point we can continue the phase to purely imaginary values $\alpha(x)\to i\hat{\beta}(x)$, so that now the non-normalizable terms become $\phi_1(x)=e^{-\hat{\beta}(x)}\psi_1(x)$ and $\overline{\phi_1}(x)=e^{\hat{\beta}(x)}\psi_1(x)$. As a consequence, $\phi$ and $\overline{\phi}$ are no longer complex conjugate of each other and we have broken Hermiticity of the theory by breaking it at the level of the boundary

values, i.e. at the level of the couplings in the dual QFT. Following the conventions of [32] we define

$$e^{\hat{\beta}(x)} = \sqrt{\frac{1+\xi(x)}{1-\xi(x)}}, \tag{2.7}$$

$$\psi_1(x) = \sqrt{1-\xi^2(x)}\,\varphi_1(x). \tag{2.8}$$

The scalar fields now asymptote to

$$\phi_1(x) = (1-\xi(x))\varphi_1(x), \tag{2.9}$$

$$\overline{\phi}_1(x) = (1+\xi(x))\varphi_1(x). \tag{2.10}$$

Notice that for $\xi(x) = 0$ we recover Hermiticity. Thus $\xi(x)$ parametrizes the non-Hermiticity of the system. Changing the sign of $\xi(x)$ simply exchanges the roles of $\phi$ and $\overline{\phi}$, thus we restrict to positive values of $\xi(x)$ only.

So far we have only considered the boundary conditions on the scalar fields. As we emphasized in the introduction, such a procedure will result in a non-unitary time evolution as soon as the parameter $\xi$ or equivalently $\beta$ becomes time dependent. A unitary time evolution can be obtained by also introducing a gauge field. In our case and generalizing to full space-time dependence this gauge field would be given by

$$A_\mu = i\partial_\mu\beta = \frac{i\partial_\mu\xi}{1-\xi^2}. \tag{2.11}$$

We will use such a gauge field to show explicitly that the time evolution in that case is equivalent to a usual time evolution with Hermitian boundary conditions obeying $\phi_1^* = \bar{\phi}_1$ where the star denotes complex conjugation.

The $\mathcal{PT}$ symmetry acts as $(r, t, x^1, x^2) \to (r, -t, -x^1, x^2)$, $(\phi, \bar{\phi}) \to (\phi, \bar{\phi})$, $A \to -A$ for the 1-form gauge field, $ds^2 \to ds^2$ and as $i \to -i$ on the imaginary unit. The non-Hermitian boundary conditions are $\mathcal{PT}$ invariant if $\xi$ is a constant. A detailed discussion of the discrete symmetries is presented in the appendix A.

For the case of constant $\xi$ this theory has been investigated in [32]. We briefly review the results now. It was found that in the regime $|\xi| < 1$ the zero temperature solutions are domain walls interpolating between two asymptotic AdS spaces. At $|\xi| = 1$ there is a critical point with the metric being exactly anti de-Sitter space whereas for $|\xi| > 1$ there are two solutions to the boundary condition which are complex conjugate to each other. In this way the holographic model reproduces the usual phase transition between the $\mathcal{PT}$-symmetric and the $\mathcal{PT}$-broken regime. The underlying reason is of course that for $|\xi| < 1$ there is always a similarity transformation (Dyson transformation) which brings the boundary conditions back to Hermitian. More surprisingly it was found that for solutions containing a black hole and thus corresponding to a finite temperature field theory solutions with real metric exists even in the $\mathcal{PT}$-broken regime. However, upon studying linear fluctuations on top of the solutions with $|\xi| > 1$ there was always an unstable mode with exponentially growing behavior. The question arises if the system would settle down to a new stable ground state upon switching on the unstable perturbation. This was also investigated in [32] but no stable ground state was found.

Our aim will be to study how the system behaves once $\xi$ is considered to be a function of time. We will generically start with an asymptotically AdS black hole solution and then vary the parameter $\xi$ as prescribed by a given function of the asymptotic time $v$. The ansatz to solve this problem depends on the radial and temporal components only. The two scalar fields are generic functions of $v$ and $u$, and the $U(1)$-connection 1-form is $A = a(v, u)dv$. The

metric fields $(f, g)$ introduced in 2.3 are also kept generic, whereas $S$ takes the simple form $S(v, u) = 1/u + \lambda(v)$ for some arbitrary function $\lambda(v)$ which is a remnant of diffeomorphism symmetry. In our numerical approach we closely follow [28]. In particular we choose the arbitrary function $\lambda(v)$ so as to keep the position of the apparent horizon, defined by the condition $\partial_v S - \frac{1}{2} u^2 f e^{-g} \partial_u S = 0$, fixed at $u_h = 1$. The precise details of the horizon fixing may be found in appendix C. Plugging the full ansatz into the equations of motion 2.2 we are left with a set of eight equations, three of which are constraints whereas the remaining five provide the dynamical evolution of the fields:

$$\frac{2g'S'}{S} - \frac{4S'}{uS} - \frac{2S''}{S} - \phi'\overline{\phi}' = 0, \tag{2.12}$$

$$f' + \frac{a'^2 S}{4S'} + f\left(\frac{S'}{S} - \frac{S}{2S'}\phi'\overline{\phi}'\right) - \frac{e^{2g}S}{2u^4 S'}\left(6 - m^2\phi\overline{\phi} - \frac{1}{2}V\phi^2\overline{\phi}^2\right) - \frac{e^g}{u^2}\left(\frac{2\dot{S}}{S} + \frac{2\dot{S}'}{S'}\right) = 0, \tag{2.13}$$

$$a'' - a'\left(g' - \frac{2S'}{S} - \frac{2}{u}\right) - iq\frac{e^g}{u^2}\left(\phi'\overline{\phi} - \phi\overline{\phi}'\right) = 0, \tag{2.14}$$

$$d\phi' + \frac{S'}{S}d\phi - iq\left(\frac{1}{2}a'\phi + \frac{aS'\phi}{S} + a\phi'\right) - u^2 f e^{-g}\frac{S'\phi'}{2S} + \frac{e^g}{2u^2}\left(m^2\phi + V\phi^2\overline{\phi}\right) + \frac{\dot{S}\phi'}{S} = 0, \tag{2.15}$$

$$d\overline{\phi}' + \frac{S'}{S}d\overline{\phi} + iq\left(\frac{1}{2}a'\overline{\phi} + \frac{aS'\overline{\phi}}{S} + a\overline{\phi}'\right) - \frac{u^2 f e^{-g}S'\overline{\phi}'}{2S} + \frac{e^g}{2u^2}\left(m^2\overline{\phi} + V\phi\overline{\phi}^2\right) + \frac{\dot{S}\overline{\phi}'}{S} = 0, \tag{2.16}$$

$$da' - a'e^{-g}dg + \frac{1}{2}u^2 e^{-g}a'\left(f' - fg' - 2f\frac{S'}{S} + 4e^g\frac{\dot{S}}{u^2 S}\right) + iq\frac{e^g}{u^2}(d\phi\overline{\phi} - d\overline{\phi}\phi) + q^2\frac{2ae^g}{u^2}\phi\overline{\phi} = 0, \tag{2.17}$$

$$\frac{e^g}{u^2}dg' - \frac{1}{2}f'' + f'\left(g' - \frac{S'}{S} - \frac{1}{u}\right) + f\left(\frac{g'S'}{S} + \frac{g'}{u} - \frac{2S'}{uS} - \frac{1}{2}g'^2 + \frac{1}{2}g'' - \frac{S''}{S}\right) + \frac{1}{4}a'^2$$
$$+ iqa\frac{e^g}{2u^2}\left(\phi'\overline{\phi} - \phi\overline{\phi}'\right) + \frac{e^g}{2u^2}\left(d\phi\overline{\phi}' + d\overline{\phi}\phi'\right) \tag{2.18}$$
$$+ \frac{e^{2g}}{2u^4}\left(6 - m^2\phi\overline{\phi} - \frac{1}{2}V\phi^2\overline{\phi}^2\right) + \frac{2\dot{S}'}{u^2 S}e^g = 0,$$

$$df + dg\left(\frac{2e^g\dot{S}}{u^2 S'} - 2f\right) + f^2 u^2 e^{-g}\left(-g' + \frac{3S}{4S'}\phi'\overline{\phi}' - \frac{lS'}{S}\right) - \frac{f'\dot{S}}{S'} - \frac{e^g}{u^2 S'}\left(d\phi d\overline{\phi}S + \ddot{S}\right)$$
$$+ f\left(-\frac{1}{2}e^{-g}\left(\frac{u^2 a'^2 S}{2S'} + u^2 f'\right) + \frac{g'\dot{S} + 4\dot{S}'}{S'} + \frac{e^g S}{2u^2 S'}\left(6 - m^2\phi\overline{\phi} - \frac{1}{2}V\phi^2\overline{\phi}^2\right) + \frac{2\dot{S}}{S}\right)$$
$$+ iqa\frac{e^g S}{u^2 S'}(d\overline{\phi}\phi - d\phi\overline{\phi}) - q^2 a^2\frac{e^g S}{u^2 S'}\phi\overline{\phi} = 0, \tag{2.19}$$

where the dot and prime stand for temporal and radial partial derivatives respectively and $d = \partial_v - \frac{1}{2}u^2 f e^{-g}\partial_u$ is the derivative along infalling null geodesics. The use of $d$ allows to get a nested system of equations. Notice that if the gauge field has trivial (or purely imaginary) boundary conditions, then the reality of the leftover fields forces $a(v, u)$ to be purely imaginary.[1] In this case we actually work with redefined gauge field $a = i\tilde{a}$ and $\tilde{a} \in \mathbb{R}$. Imposing AAdS boundary conditions and recalling that $S(v, u) = 1/u + \lambda(v)$ we find the asymptotic

---

[1] A similar fact was observed in the field theoretical model of [22].

solution to be

$$f(v,u) \simeq \left(\frac{1}{u} + \lambda\right)^2 - 2\dot{\lambda} + f_1 u + \mathcal{O}(u^2),$$

$$g(v,u) \simeq -\frac{1}{4}(1-\xi^2)\varphi_1^2 u^2 - \frac{1}{3}\left(\frac{1}{2}(1-\xi^2)\lambda\varphi_1 + (1+\xi)\phi_2 + (1-\xi)\overline{\phi}_2\right)u^3 + \mathcal{O}(u^4),$$

$$\phi(v,u) \simeq u(1-\xi)\varphi_1 + u^2\phi_2 + \mathcal{O}(u^3),$$

$$\overline{\phi}(v,u) \simeq u(1+\xi)\varphi_1 + u^2\overline{\phi}_2 + \mathcal{O}(u^3),$$

$$a(v,u) \simeq a_0 + a_1 u + \left(\frac{1}{2}iq\varphi_1\left((1+\xi)\phi_2 - (1-\xi)\overline{\phi}_2\right) - a_1\lambda\right)u^2 + \mathcal{O}(u^3),$$

(2.20)

where all free coefficients are taken to be functions of time. The subleading terms in $a$ and $f$ are further constrained by the equations of motion to obey

$$\dot{a}_1 = iq\left(2\varphi_1\dot{\xi} + (1+\xi)\phi_2 - (1-\xi)\overline{\phi}_2\right)\varphi_1 - 2q^2 a_0(1-\xi^2)\varphi_1,$$

$$\dot{f}_1 = -\partial_v\left[(1+\xi)\varphi_1\right]\partial_v\left[(1-\xi)\varphi_1\right] - \frac{1}{6}\left[(1+\xi)\varphi_1\right]^3\partial_v\left(\frac{\phi_2}{(1+\xi)^2\varphi_1^2}\right)$$

$$\quad - \frac{1}{6}\left[(1-\xi)\varphi_1\right]^3\partial_v\left(\frac{\overline{\phi}_2}{(1-\xi)^2\varphi_1^2}\right) + \frac{1}{6}\lambda^3\partial_v\left[\frac{1}{\lambda^2}(1-\xi^2)\varphi_1^2\right]$$

$$\quad + iqa_0\varphi_1(4\varphi_1\dot{\xi} + (1+\xi)\phi_2 - (1-\xi)\overline{\phi}_2) - q^2 a_0^2(1-\xi^2)\varphi_1^2.$$

(2.21)

The subleading free coefficients in the expansion are directly related to the 1-point functions in the dual quantum field theory. Making use of the holographic prescription for the renormalized action 2.1 and plugging in the expansion 2.20 we obtain the expectation values of the various operators in the dual field theory as

$$2\kappa^2\langle\mathcal{O}_\phi\rangle = \overline{\phi}_2 - \dot{\xi}\varphi_1 - (1+\xi)\dot{\varphi}_1 - iqa_0(1+\xi)\varphi_1,$$

$$2\kappa^2\langle\mathcal{O}_{\overline{\phi}}\rangle = \phi_2 + \dot{\xi}\varphi_1 - (1-\xi)\dot{\varphi}_1 + iqa_0(1-\xi)\varphi_1,$$

$$2\kappa^2\langle J^v\rangle = -a_1 := -i\tilde{a}_1,$$

$$\kappa^2\langle T_{vv}\rangle = -f_1 - \frac{1}{6}\varphi_1\left((1-\xi)\overline{\phi}_2 + (1+\xi)\phi_2\right),$$

$$\kappa^2\langle T_{x_1x_1}\rangle = \kappa^2\langle T_{x_2x_2}\rangle = -\frac{1}{2}f_1 + \frac{1}{6}\varphi_1\left((1-\xi)\overline{\phi}_2 + (1+\xi)\phi_2 + 3\xi\dot{\xi}\varphi_1\right).$$

(2.22)

Note in particular that the expectation value for the charge $\langle J^v\rangle$ is purely imaginary. This is a clear sign for the non-Hermiticity of the time evolution. As we will see at late times the system settles down to equilibrium solutions in the $\mathcal{PT}$ symmetric regime with vanishing charge. The details of the renormalization can be found in B.

# 3 Numerical Results

In this section we show explicitly the temporal evolution of the system for some concrete interesting examples. Both initial and final states correspond to equilibrium unless otherwise stated. We first study the response of the system whenever $\xi$ interpolates between two different values, starting at the Hermitian point $\xi = 0$ and ending either in the $\mathcal{PT}$-symmetric phase, i.e. $|\xi| < 1$, or at the exceptional point. Afterwards we shall dive for some finite time into the $\mathcal{PT}$-broken, i.e. $|\xi| > 1$. Emphasis should be given to the fact that no equilibrium solution exists for

the $\mathcal{PT}$-broken regime at zero temperature, whereas the real solutions at finite temperature where found to be unstable. Motivated by the results of the previous cases, we also take a look at the situation where $\xi$ is forever oscillating. Afterwards we monitor the system under a quench in the Hermitian direction. Finally, we discuss the mapping from a non-Hermitian evolution to a genuinely Hermitian one. All simulations are made for $V = 3$ and $q = 1$. Further details regarding the numerical setup may be found on the appendix C.

## 3.1 $\mathcal{PT}$-symmetric evolution

In figures 1 and 2 we display the evolution of the quantum field theory observables when the $\mathcal{PT}$-breaking parameter $\xi$ evolves according to

$$\xi(v) = \xi_i + \frac{\xi_f - \xi_i}{2}\left[1 + \tanh\left(\frac{v - v_m}{\tau}\right)\right],\tag{3.1}$$

for different values of $\tau$. Both groups of simulations start at the Hermitian point $\xi_i = 0$, the difference being that in 1 we end up in the $\mathcal{PT}$-symmetric regime, in particular $\xi_f = 0.8$, whereas in 2 we land in the exceptional point, i.e. $\xi_f = 1$.

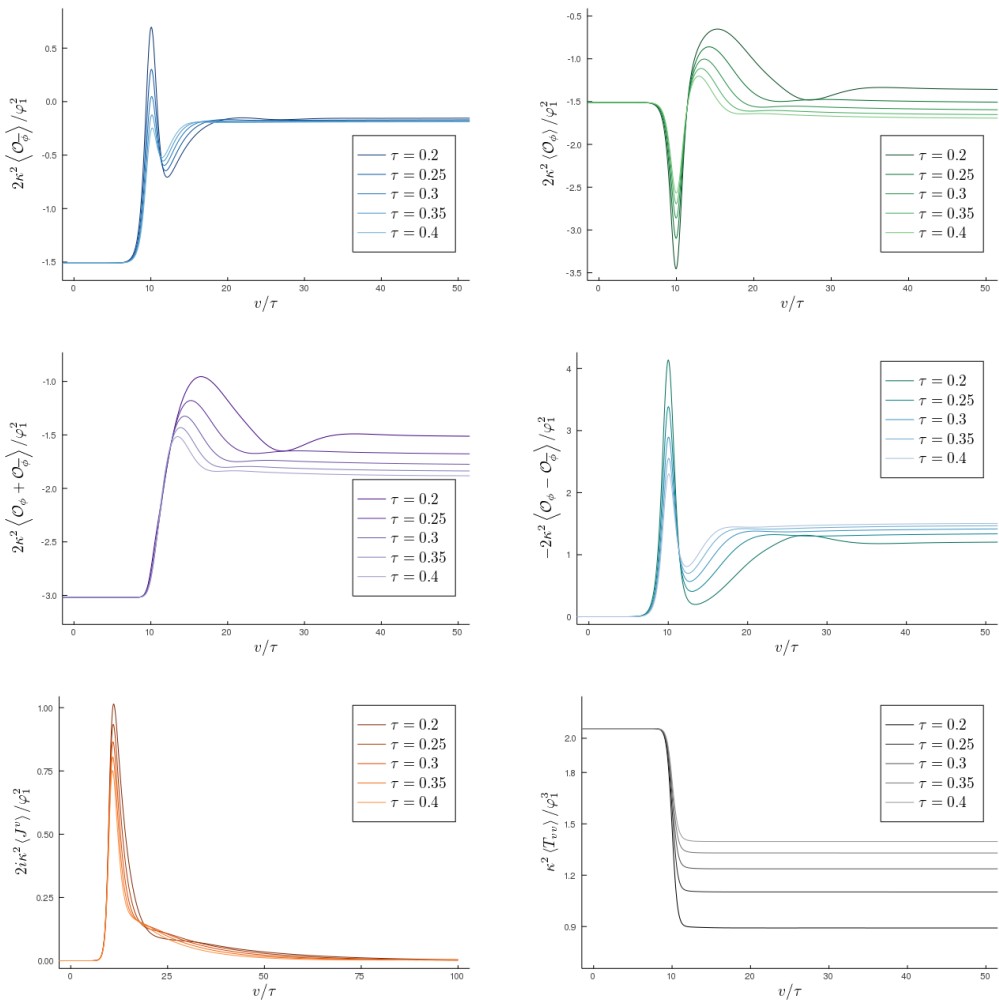

Figure 1: Expectation values 2.22 for a quench with profile 3.1 for several values of $\tau$ which interpolates between the Hermitian point $\xi_i = 0$ and a final value $\xi_f = 0.8$. We set $v_m = 10\tau$.

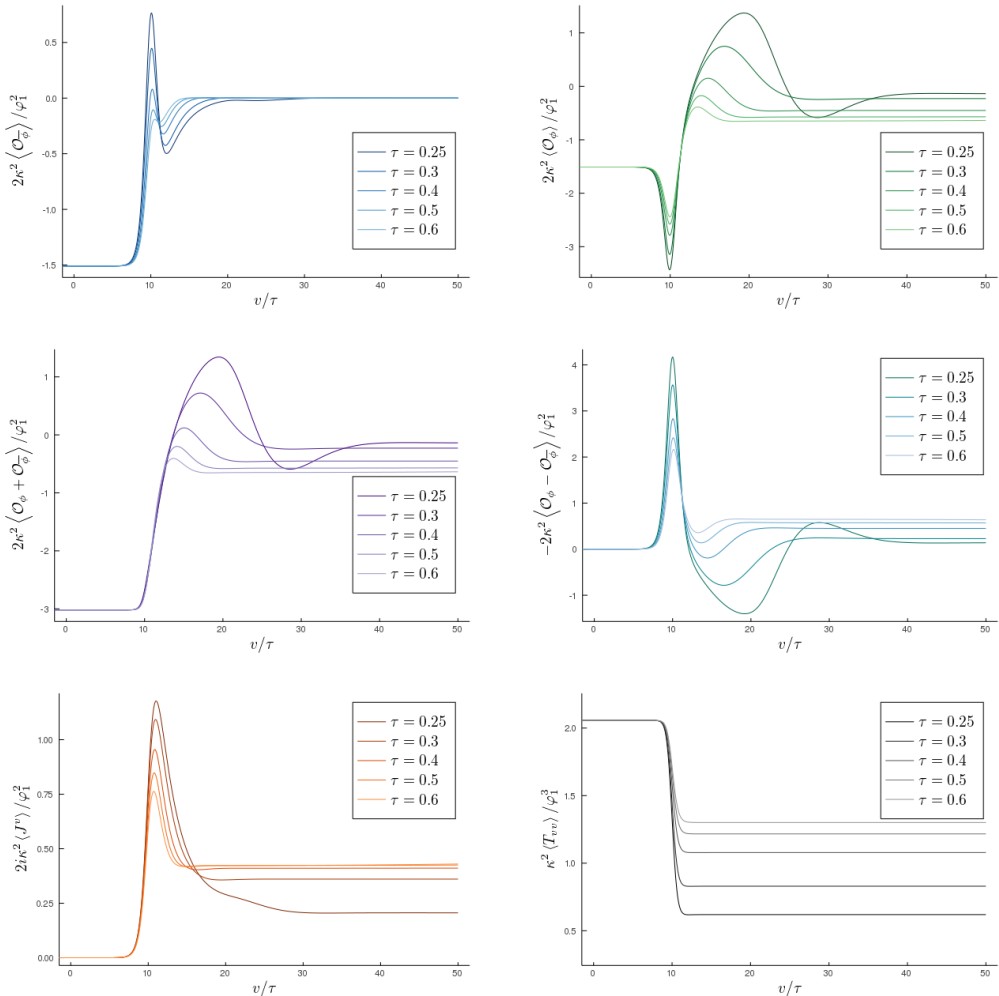

Figure 2: Expectation values 2.22 for a quench with profile 3.1 for several values of $\tau$ which interpolate between the Hermitian point $\xi_i = 0$ and the exceptional point $\xi_f = 1.0$. We set $v_m = 10\tau$. Note that the current settles to a final equilibrium value different from 0. This feature is distinctive for the exceptional point.

A few remarks are in order. First of all note that in all cases the system settles down to a new equilibrium state. However, the behaviour of the current $\langle J^v \rangle$ depends on whether one ends up in $\mathcal{PT}$-symmetric point or in the exceptional point. In the first case it relaxes to zero. The system reaches at late times the equilibrium solution already described in [32], where no gauge field was switched on. In the second case, in which one ends up at the exceptional point $\xi = 1$, the expectation value of the current settles down to a non-trivial value, making it into a solution not previously described. We emphasize that this is still a purely imaginary expectation value of the charge operator. The exceptional point can be reached from a Hermitian Hamiltonian only by taking a limiting procedure. There is no unique map back to a Hermitian theory. Our result also suggests that the properties of the theory at the exceptional point depend on how this point is reached.

As for the scalar operators, they follow the expected evolution from the initial to the final equilibrium states. An extra piece of information is provided when plotting the sum and the difference of both scalar operators. They are identified as the Hermitian and anti-Hermitian parts of the expectation values respectively. The fact that we start the evolution in the Hermitian point translates into the non-Hermitian operator vanishing initially. Note that the expectation

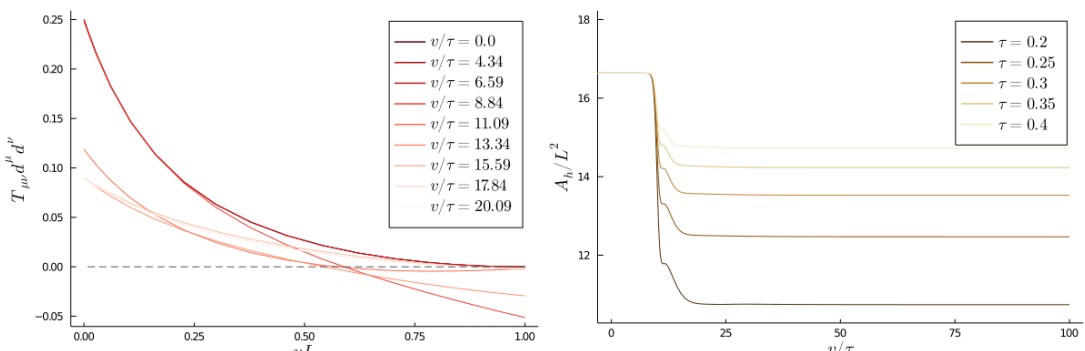

Figure 3: (Top) We check the NEC $T_{\mu\nu}d^\mu d^\nu \geq 0$ for the particular simulation of figure 1 with $\tau = 0.4$ at different stages $v/\tau$ of the evolution. It is clearly violated, especially at the horizon $uL = 1$. (Bottom) We display the size of the apparent horizon for the simulations of figure 1. All of them show a shrinking horizon, which is tightly related to violating the NEC.

value for the Hermitian operator also develops a non-trivial profile despite the quench being made only in the non-Hermitian direction.[2]

Secondly, it is interesting to observe that the smaller the value of $\tau$, the lower the energy of the dual quantum field theory $\kappa^2 \langle T_{\nu\nu} \rangle$. Such decrease in the energy is encompassed with a decrease in temperature and a subsequent decrease of the area of the black hole apparent horizon (see for instance figure 3 *bottom*), which is simply given by $4\pi(1+\lambda(v))^2$. It should be expected that a subset of all possible profiles $\xi(v)$ result into a dramatic shrink of the horizon, so that one dangerously approaches the singularity inside of it. Hence gravity is no longer weakly coupled there and the effective description breaks down. Such a construction, for a specific profile $\xi(v)$, is shown in section 3.3.

The shrinking of the horizon poses no contradiction to the second law of black hole thermodynamics, which states that the area of the horizon has to grow in any given process provided the null energy condition (NEC) is satisfied. Indeed, all the non-Hermitian processes here studied do violate the NEC. It suffices to take the null vector tangent to infalling null geodesics $d$ defined after 2.19. Then $T_{\mu\nu}d^\mu d^\nu = (d\phi - iqa\phi)(d\overline{\phi} + iqa\overline{\phi})$, where $T_{\mu\nu}$ is the bulk energy momentum tensor, should be positive if the NEC was to hold. We have monitored this during the time evolution and found that the NEC is violated. The same applies for sections 3.2 and 3.3. In figure 3 *top* we display the quantity $T_{\mu\nu}d^\mu d^\nu$ at different stages of the evolution for the particular simulation of figure 1 with $\tau = 0.4$. The NEC is known to hold for standard forms of matter, so in this respect it appears that the non-Hermitian extension of the holographic model 2.1 yields to *exotic* matter in the bulk. Nonetheless, the bulk is here regarded as an effective description of the quantum field theory. It is the QFT that is taken to be fundamental and it should not be worrisome to find uncommon features in the bulk.

## 3.2 $\mathcal{PT}$-broken evolution

It is well known that in the $\mathcal{PT}$-broken phase, the time-independent Hamiltonian of the theory can no longer be mapped to a Hermitian one. However, when the $\mathcal{PT}$-breaking parameter is promoted to be a function of the coordinates, in particular of time, one can wonder whether

---

[2]Recalling the parametrisation 2.9 and 2.10 of the boundary values for the scalar fields it is clear that varying only $\xi$ amounts to varying the difference $\phi_1 - \overline{\phi}_1$ while keeping the corresponding sum fixed. Thus the quench is performed in the non-Hermitian direction. The complementary approach of quenching only in the Hermitian direction is studied in section 3.4.

a solution exists after one spends a finite amount of time in the $\mathcal{PT}$-broken region. In order to address this question we give $\xi$ the profile

$$\xi(v) = \xi_i + \frac{\xi_m - \xi_i}{2}\left[1 + \tanh\left(\frac{v - v_m}{\tau}\right)\right] + \frac{\xi_f - \xi_m}{2}\left[1 + \tanh\left(\frac{v - \tilde{v}_m}{\tilde{\tau}}\right)\right]. \qquad (3.2)$$

We will focus on the evolution that takes place when both the initial and final states lie in the Hermitian point, i.e. $\xi_i = \xi_f = 0$. In figure 4 we present several profiles for the $\mathcal{PT}$-breaking parameter as well as the associated observables. Surprisingly one can enter and leave the $\mathcal{PT}$-broken regime and still find a suitable final equilibrium state. However we find restrictions to the maximum value $\xi$ can take along the evolution. In particular, if one goes above some critical value $\xi_c$, no real solution is found. Such critical value depends on the particular state one has at each time, making its quantitative prediction unfeasible. A similar feature was described in [32] for the time independent case.

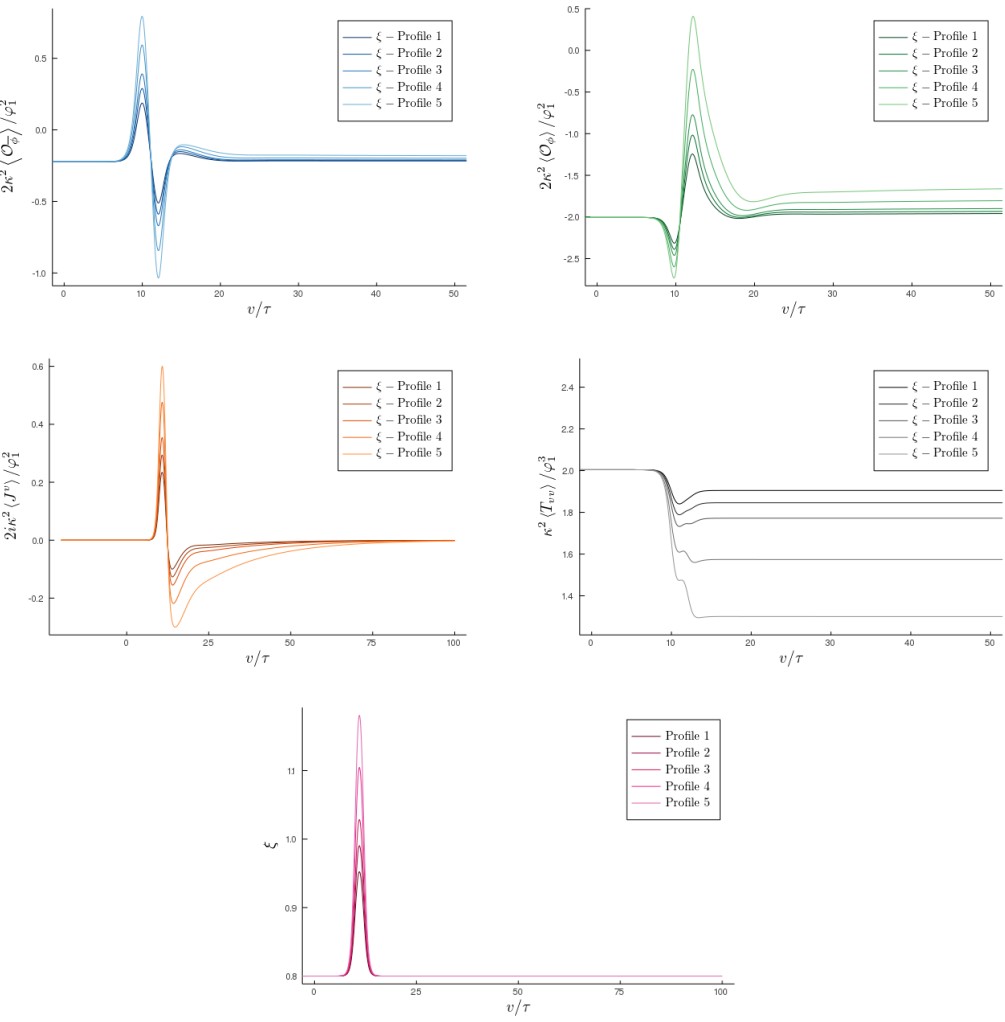

Figure 4: Expectation values 2.22 for a quench with profile 3.2 for $\xi_m = \{1.0, 1.05, 1.1, 1.2, 1.3\}$. The initial and final states are located at $\xi = 0.8$ and we set $\tau = \tilde{\tau} = 0.25$, $v_m = 10\tau$ and $\tilde{v}_m = 12\tilde{\tau}$. We explore here the $\mathcal{PT}$-broken regime $|\xi| > 1$ going deeper and deeper into it. Even though the system spends significant amount of time in the $\mathcal{PT}$ broken regime it settles down to real and well behaved solutions once it re-enters the $\mathcal{PT}$ symmetric regime.

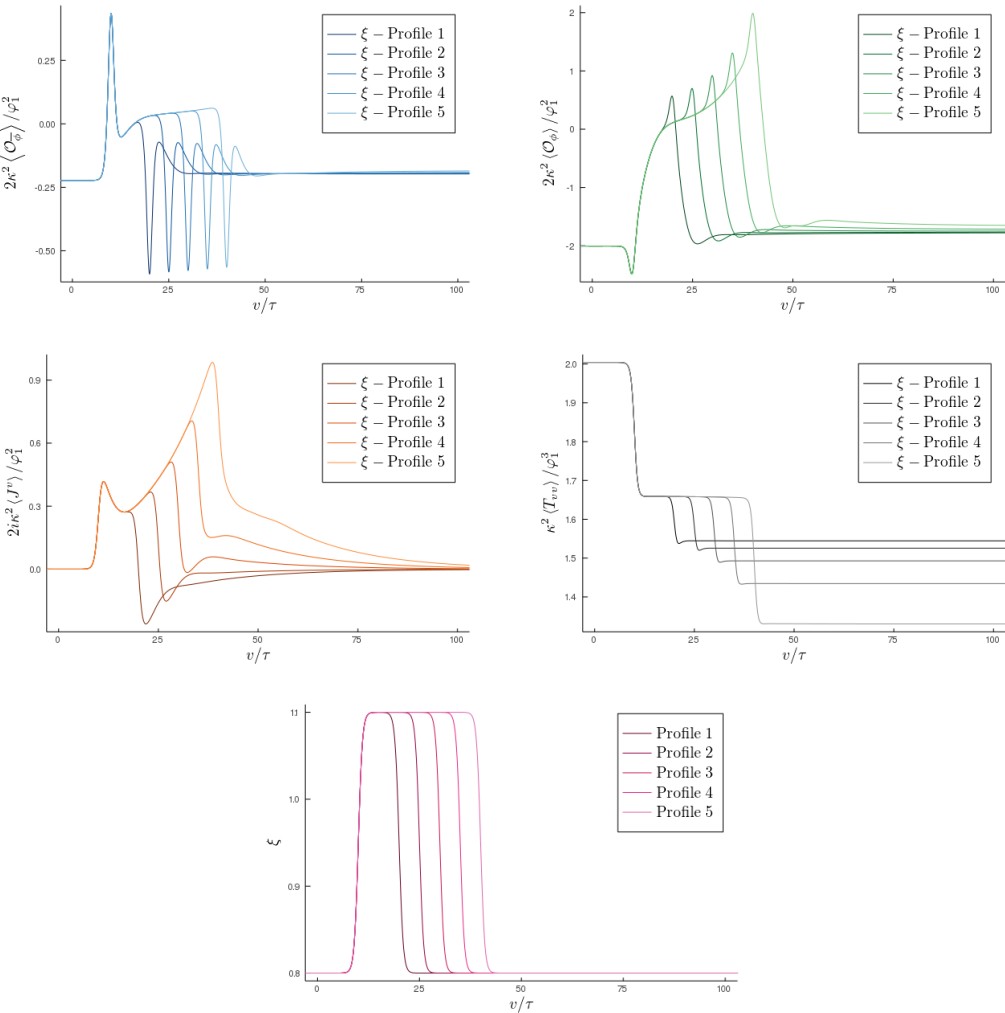

Figure 5: Expectation values 2.22 for a quench with profile 3.2 for $\tilde{v}_m/\tilde{\tau} = \{20, 25, 30, 35, 40\}$. The initial and final states are located at $\xi = 0.8$, besides we set $\xi_m = 1.1$ and $\tau = \tilde{\tau} = 0.25$. We explore here the $\mathcal{PT}$-broken regime $|\xi| > 1$ remaining progresively more time into it. An instability starts to develop, see for instance the green and orange curves, but it eventually fades out as we re-enter the $\mathcal{PT}$-symmetric region.

At this point we already know that we can dive into the $\mathcal{PT}$-broken regime. Apparently one cannot stay forever there, as these solutions are indeed unstable [32], yet one could in principle stay some arbitrarily long time there provided it eventually goes back into the $\mathcal{PT}$-symmetric region. In figure 5 we show how these kind of constructions look like. The instability is crearly there, as soon as $\xi > 1$ both scalar operators and the current expectation values start to diverge, yet once we re-enter $|\xi| < 1$ the system relaxes down to an appropriate equilibrium state.

## 3.3 Oscillating evolution

All the features displayed so far, particularly the fact that the apparent horizon shrinks, suggest that one cannot have a situation where the $\mathcal{PT}$-breaking parameter is always varying with time even within the $\mathcal{PT}$-symmetric region. To illustrate this fact we study the evolution of

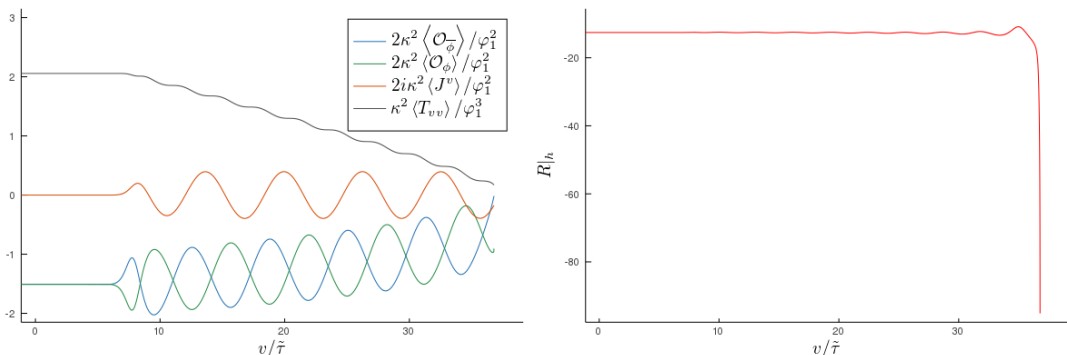

Figure 6: (Top) Expectation values 2.22 for a quench with profile 3.3 and parameters $\xi_i = 0$, $A = 0.2$, $\omega = 1/(4\pi)$, $v_m = 2.5$ and $\tau = 0.25$. This corresponds to non-Hermitian oscillations around the Hermitian point. (Bottom) Evolution of the Ricci scalar at the apparent horizon for the same simulation as in *top*. It diverges at the end as a consequence of the shrinking of the apparent horizon, which gets us too close to the interior singularity.

the system when $\xi$ is given the profile

$$\xi = \xi_i + A\sin(\omega v)\left[1 + \tanh\left(\frac{v - v_m}{\tau}\right)\right]. \tag{3.3}$$

In particular we study small oscillations around the Hermitian point. This can be thought of as mimicking non-Hermitian fluctuations of an otherwise Hermitian coupling in the dual quantum field theory. The result is shown in figure 6. Indeed the horizon diminishes its area and at some point one gets too close to the singularity. An explicit evaluation of the Ricci scalar at the horizon reveals divergent behaviour, confirming thus the previous statement. As a consequence, one would need to take quantum gravity effects into account in order to carry on with the time evolution.

## 3.4 Quenching the Hermitian direction

In previous sections we have studied the response of the system as we quench the parameter $\xi$ controlling the non-Hermiticity of the system, i.e. we quench in the non-Hermitian direction. In this section we envisage a different problem in which we start with non-Hermitian boundary conditions but we perform a quench in the Hermitian direction. Specifically we parametrize the boundary conditions now as

$$\phi_1(x) = (\chi(x) - 1)\zeta_1(x), \tag{3.4}$$

$$\overline{\phi}_1(x) = (\chi(x) + 1)\zeta_1(x), \tag{3.5}$$

so that the quench so far described corresponds to varying $\chi(t)$ with time while keeping $\zeta_1$ fixed. One can recover the original parametrisation by sending $\zeta_1(x) \to \xi(x)\varphi_1(x)$ and $\chi(x) \to 1/\xi(x)$. Note that the $\mathcal{PT}$-symmetric regime lies now within $|\chi| > 1$, with the Hermitian point located at infinity and the exceptional point kept at $\chi_c = 1$.

Once again we focus on an interpolating profile for $\chi(t)$ as given in 3.1. We interpolate between $\chi_i = 10$ and $\chi_f = 2.0$ for several values of $\tau$. The temporal evolution of the expectation values 2.22 is depicted in figure 7. The response on both scalar expectation values are fairly similar. However, their difference, which as explained in section 3.1, gives the expectation value for the non-Hermitian operator, does develop a non-trivial profile. The current operator

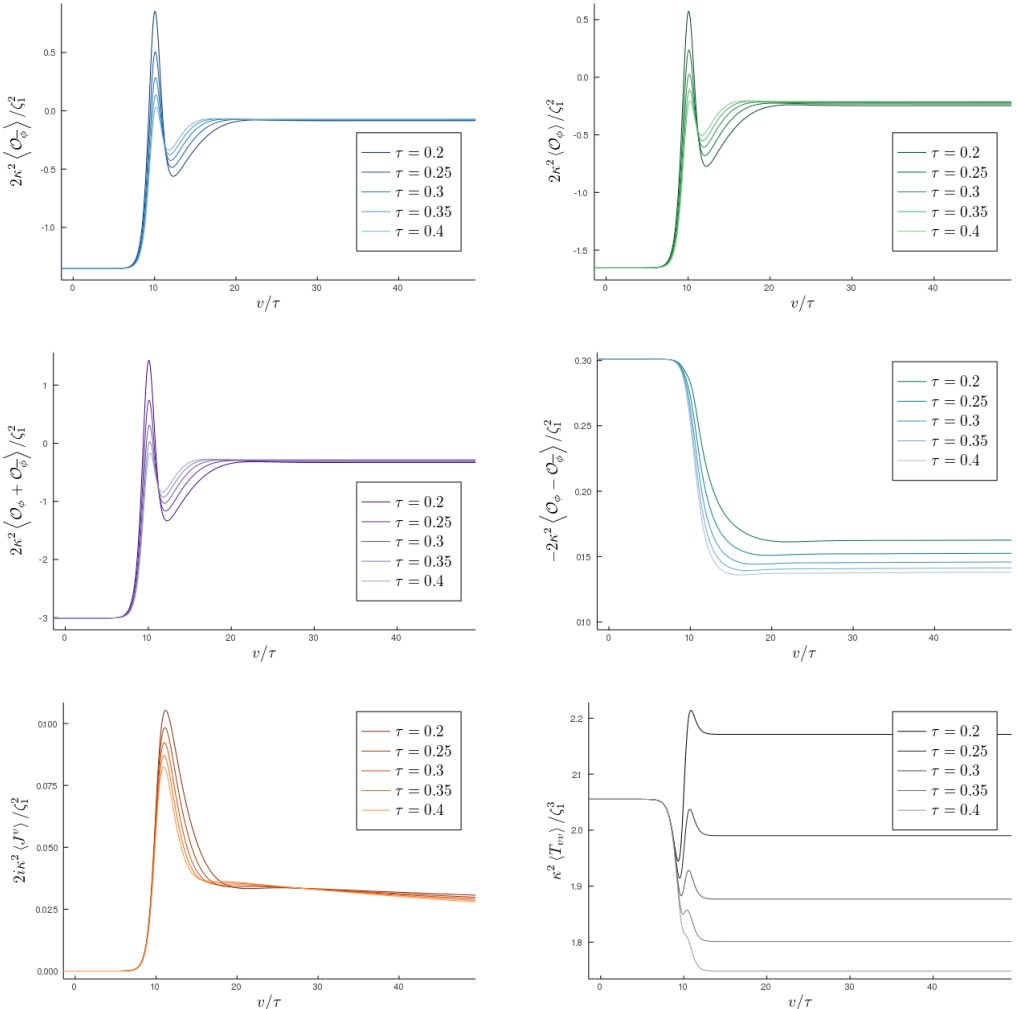

Figure 7: Expectation values 2.22 for a quench with parametrisation 3.4 of the boundary values of the fields with profile 3.1 for $\chi$ and for several values of $\tau$ labelling different curves. We interpolate between $\chi_i = 10$ and $\chi_f = 2.0$, and set $v_m/\tau = 10$.

however takes considerably more time to settle down to equilibrium (where it should vanish) as compared with, for instance, the quench in figure 1.

Another distinctive feature is that now the energy $\langle T_{vv} \rangle$ can be either increasing or decreasing. Very interestingly we find that quenching the Hermitian direction results now into a growing apparent horizon whatsoever, as can be appreciated in figure 8 *bottom* where we display how the area of the apparent horizon evolves with time. Nevertheless the null energy condition is still being violated as a consequence of the non-Hermitian boundary condition. A slight violation is observed taking the same null vector $d$ as in section 3.1 or more clearly taking the null vector $k = \partial_u$, which yields $k^\mu k^\nu T_{\mu\nu} = \phi'\overline{\phi}'$. Such quantity is exhibited at different stages of the evolution in figure 8 *top*, corresponding to the simulation in figure 7 with $\tau = 0.2$. Indeed $k^\mu k^\nu T_{\mu\nu}$ becomes negative within the bulk, violating thus the null energy condition. Either with $d$ or with $k$ the NEC is obeyed at the horizon. This fact is relevant to explain the growth of the horizon as we proceed to discuss.

The mathematical reason behind the fact that in section 3.1 the violation of the NEC gave a decreasing horizon and not here can be understood form equations C.2 and C.3 in appendix

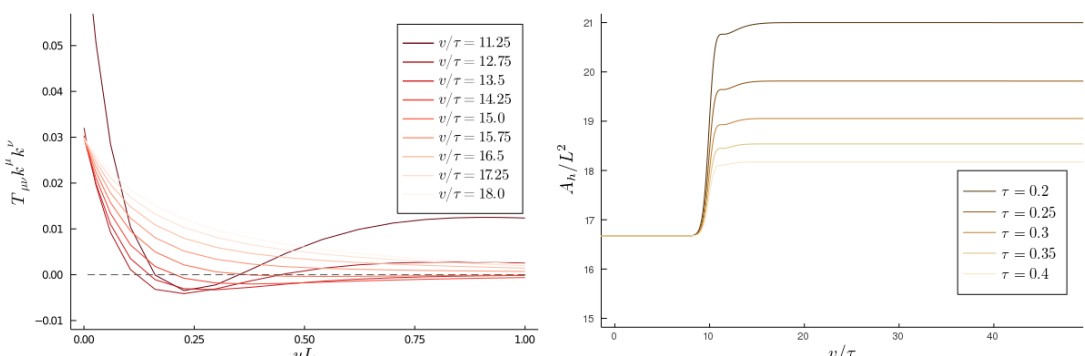

Figure 8: (Top) We check the NEC $T_{\mu\nu}k^{\mu}k^{\nu} \geq 0$ for the particular simulation of figure 7 with $\tau = 0.2$ at different stages $v/\tau$ of the evolution. It is violated in between the boundary and the horizon. The polygonal form of the curves is a consequence of using only 20 gridpoints in the radial direction, as explained in appendix C. (Bottom) We display the size of the apparent horizon for the simulations of figure 7. All of them show a growing horizon.

C. First recall that the area of the apparent horizon is given by $4\pi(1 + \lambda(v))^2$, so its evolution is controlled solely by the radial shift function $\lambda$. According to equation C.2 the sign of $\dot{\lambda}$ is the opposite to the sign of $f$ at the horizon ($u_h = 1$), wich in agreement with C.3 is proportional to the sign of $(d\phi - iqa\phi)(d\overline{\phi} + iqa\overline{\phi})$ at the horizon. Now this last equation is precisely the null energy condition for the vector $d = \partial_v - \frac{1}{2}u^2 f e^{-g} \partial_u$, so the growth or decrease of the area of the apparent horizon depends on whether $d^{\mu}d^{\nu}T_{\mu\nu} \geq 0$ is violated at the horizon or not.

Finally it is worth noting that a forever oscillating quench in the Hermitian direction would not suffer from the pathology found in section 3.3, where the quench was performed in the non-Hermitian direction.

## 3.5  Mapping back to Hermiticity

As we discussed in the introduction, a unitary time evolution may be obtained by switching on an additional non-Hermtitian gauge field. In holography this enters as the leading mode in the asymptotic expansion (2.20) of the gauge field by setting

$$a_0 = \frac{i\partial_v\xi}{1-\xi}, \tag{3.6}$$

which compensates for the time dependence of $\xi$ in the boundary values of both scalar fields.

We provide now an explicit realisation of this setup. In particular we repeat the simulation of figure 1 with $\tau = 0.4$ and the non-normalizable mode of the gauge field switched on. We shall refer to this first quench as simulation A. Such simulation should be equivalent to a genuinely Hermitian quench achieved by sending $\xi \to 0$, so that we sit on the Hermitian point, and vary the boundary value of the scalar fields according to $\varphi_1(v) = \sqrt{1 - \theta(v)^2}\varphi_1(0)$, where $\theta(v)$ follows the same profile that $\xi(v)$ took on simulation A. We name this second quench as simulation B.

It is easy to check that 3.6 already gives a solution for the gauge field equation of motion. Consequently, simulations A and B are truly related by a (complexified) gauge transformation and we should thus have $\phi_A = e^{\beta}\phi_B$ and $\overline{\phi}_A = e^{-\beta}\overline{\phi}_B$ with $\beta$ defined as in 2.7 and the subscript labelling the corresponding simulation. In figure 9 we display the subleading modes of both scalar fields for the Hermitian simulation B and for the simulation A once we apply the gauge transformation described before. This way it is manifest that both quenches are

phisically equivalent. Furthermore we also remark that in this case because of the singularity in the gauge field at $\xi = 1$ it is not possible to go into the $\mathcal{PT}$ broken regime or even to reach the exceptional point.

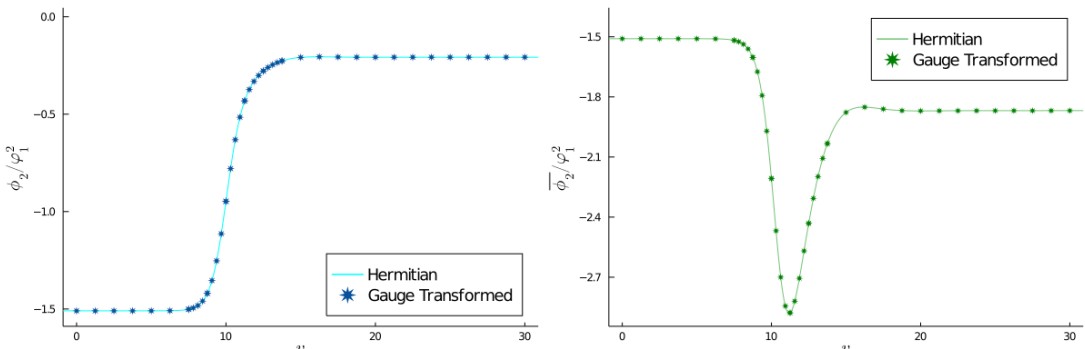

Figure 9: Subleading coefficients of both scalar fields as defined in 2.20 for simulations B, labelled as Hermitian, and for simulation A once we perform the gauge transformation, labeled Gauge Transformed. Both simulations overlap and are phisically equivalent.

## 4  Conclusions

We have studied non-Hermitian holographic quantum quenches and have found a quite rich phenomenology. As we have outlined in the introductory section, the question of how to define even the time evolution arises with two distinct possibilities, namely a non-unitary one vs. a unitary time evolution with introduction of a non-Hermitian gauge field. Holography allowed us to study both possibilities rather easily. It turned out that the non-Hermitian time evolution violates the Null Energy Condition in the bulk of the AdS spacetime. Interestingly there is a difference if one quenches purely in the non-Hermitian direction. In this case the NEC is violated at the Horizon and this leads to a shrinkage of the Horizon and eventually the appearance of divergence in the curvature signaling the possible appearance of a naked singularity. We also found that one can make excursions of finite duration into the $\mathcal{PT}$ broken regime. Furthermore we found that if one ends precisely at the $\mathcal{PT}$ critical point some finite non-zero and purely imaginary charge $\langle J^\nu \rangle$ is induced.

On the other hand a quench in the Hermitian direction in the presence of a constant non-Hermitian coupling still leads to a non-unitary time evolution but this time the NEC is violated in the bulk but not on the Horizon. This leads to a more standard time evolution with growing Horizon. This could perhaps have been expected since in that case one only varies the coupling of a perfectly nice Hermitian operator.

Finally we have explicitly demonstrated that upon introducing a non-Hermitian gauge field the time evolutions is exactly equivalent to a standard Hermitian quench.

The numerical setup and its interpretation rely on the concept of the apparent horizon. However it is the event horizon the one that ultimately isolates the black hole interior. The event horizon is teleological in nature and the full evolution of spacetime must be known in order to locate it accurately. In standard general relativity simulations the NEC is satisfied and the apparent horizon is guaranteed to lie inside the event horizon [34]. Consequently, integrating up to the apparent horizon one is including all physically relevant information which may influence the region outside the event horizon. Intuitively, a light ray outside the

apparent horizon at some time $v_*$ will tend to move outside of it and eventually reach the boundary at infinity provided the light ray is also outside of the event horizon. It may be the case that the light ray is outside the apparent horizon but inside the event horizon. Then it will start moving towards infinity, but the apparent horizon grows faster than the light ray, forcing it to change direction and eventually reach the singularity.

On the other hand, our non-Hermitian simulations violate the NEC and there is no robust statement regarding the relative position of the apparent and event horizons. We can exploit the intuition gained above and the fact that the evolution of the apparent horizon (either shrinkage or expansion) is monotonous (see figures 3 and 8) to infer where the event horizon should be. Let us focus on the monotonous shrinking. Suppose we have a lightray slightly outside the apparent horizon at some time $v_*$. Then it propagates towards infinity. Subsequently, the apparent horizon will further contract and nothing keeps the light ray from reaching infinity. On the contrary, a light ray starting inside the apparent horizon at some time $v_*$ will tend to move towards the singularity. As before, if the apparent horizon shrinks faster than the light ray falls, it will be forced to reverse direction and eventually reach infinity, showing that the event horizon is actually further inside the apparent horizon. This kind of behaviour has been shown explicitly using the Vaidya metric as a toy model [35].

The fact that we are integrating up to the apparent horizon has now several consequences. Firstly, the region in between the apparent and event horizons is now causally connected with infinity and has not been included in the simulation. The effect this has in the boundary is likely to be small, especially at early times, but a more careful analysis is needed. Secondly, locating the actual event horizon is involved in this setup, for one would need to know the metric inside the apparent horizon. Finally, further quenching the non-Hermitian direction as in section 3.3 would result in the absence of the event horizon, suggesting that a reliable simulation should include the full interior of the apparent black hole.

While our studies were limited to holographic field theories we think they contain valuable lessons also for weakly coupled quantum field theories. In particular we expect that also in weakly coupled field theories there will be a noticeable difference in the time evolution depending on whether the non-Hermitian or the Hermitian coupling is varied. We leave these intriguing questions for future study.

## Acknowledgments

We thank M. Chernodub and P. Millington for discussions. This work is supported by Grant CENTRO DE EXCELENCIA "SEVERO OCHOA", CEX2020-001007-S funded by MCIN/AEI/10.13039/501100011033, by grants PGC2018-095976-B-C21 and PGC2018-095862-B-C21 funded by MCIN/AEI/10.13039/501100011033 and by ERDF "A way of making Europe". SMT is supported by an FPI-UAM predoctoral fellowship.

## A    Discrete symmetries in the holgoraphic model

We discuss here how the discrete symmetries parity $P$, time reversal $T$ and charge conjugation $C$ act in the holographic model. It is easiest to switch to form language in which the gauge field is a 1-from $A = A_\mu dx^\mu$. Similarly instead of a metric we use a vielbein 1-form with $ds^2 = e^a e^b \eta_{ab}$. Then the spin connection is defined by the condition of vanishing torsion $de^a + \omega^a{}_b e^b = 0$ and the curvature 2-form is $R^{ab} = d\omega^{ab} + \omega^a{}_c \wedge \omega^{cb}$. The action can be

written as

$$S = \int \left[ R^{ab} \wedge *(e^c \wedge e^d) \eta_{ac} \eta_{bd} + \bar{D}\bar{\phi} \wedge *(D\phi) + dA \wedge *(dA) + V(\bar{\phi}\phi) *(1) \right]. \qquad \text{(A.1)}$$

Here $*$ denotes the Hodge dual taking a $p$-form to a $D-p$ form in $D$ space-time dimensions, in particular $*(1)$ is the volume $D$-form. In order to define parity and time reversal we assume that the action is integrated on a manifold of the form $\mathbb{R} \times \mathcal{M}_{D-1}$ where the time $t$ parametrizes the $\mathbb{R}$ factor. Furthermore $\mathcal{M}_{D-1}$ is taken to be orientable. This is indeed the case for the asymptotically AdS spaces in our model.

For concreteness we specialize now to coordinates $t, r, x^1, x^2$ where $t$ parametrizes the $\mathbb{R}$ factor and parity (orientation reversal) on $\mathbb{M}_3$ is implemented by $P: x^1 \to -x^1$. We remind the reader that a definition of parity that works in all space time dimension is by reflection of one (space-like) coordinate. Since we want this to descend onto the parity transformation in the dual field theory living on the boundary at $r = \infty$ and parametrized by $(t, x^1, x^2)$. In order to get a homogeneous transformation law for the gauge field we take $P: A \to A$. In particular this implies that the electric field transforms in same way as the coordinates, i.e. $E_1 \to -E_1$. Now let us infer the transformation of the scalar field. We can do this by noting $D\phi = d\phi - iqA\phi$. The exterior derivative is invariant $P: d \to d$. Therefore we have $P: (\phi, \bar{\phi}) \to (\phi, \bar{\phi})$ for the scalars. The vielbein is taken to be parity even $P: e^a \to e^a$ which entails the spin connection and the curvature to be parity even as well. To see invariance of the action we note that the integral is parity odd and that the Hodge star is also parity odd. Then the action is parity invariant and parity is a symmetry of the theory.

In order not to clutter the expressions too much we have suppressed the arguments of the fields. We note that the transformed fields have to be evaluated at the parity reflected point, i.e. $P: \phi(r, t, x^1, x^2) \to \phi(r, t, -x^1, x^2)$ and so on. The analogous statement holds for the time reversal transformation $T$.

Time reversal $T: t \to -t$ is anti-linear and also acts as $i \to -i$ on the imaginary unit. Its action on the field can be defined by $T: A \to -A$ which makes the field strength 2-form T-odd and thus the electric field $E_i = F_{ti}$ T-even. Analyzing the covariant derivative we have $T(D\phi) = dT(\phi) + i(-A)T(\phi)$ which gives $T: (\phi, \bar{\phi}) \to (\phi, \bar{\phi})$. The vielbein, spin-connection and curvature are $T$-even. Again the action is invariant under $T$.

We can also define a charge conjugation transformation by $C: A \to -A$ and $C: (\phi, \bar{\phi}) \to (\bar{\phi}, \phi)$ without any action on the coordinates and $i$.

We summarize the transformation laws in the following table

| | $A$ | $\phi$ | $\bar{\phi}$ | $ds^2$ | $i$ | $(r, t, x^1, x^2)$ |
|---|---|---|---|---|---|---|
| P | $A$ | $\phi$ | $\bar{\phi}$ | $ds^2$ | $i$ | $(r, t, -x^1, x^2)$ |
| T | $-A$ | $\phi$ | $\bar{\phi}$ | $ds^2$ | $-i$ | $(r, -t, x^1, x^2)$ |
| C | $-A$ | $\bar{\phi}$ | $\phi$ | $ds^2$ | $i$ | $(r, t, x^1, x^2)$ |

The definition time reversal symmetry is not unique. Its defining property is reflection of the time coordinate and complex conjugation, but that does not uniquely determine $T$. We could also define an alternative time reversal transformation by taking the product $T' = CT$ acting as $T': (r, t, x^1, x^2) \to (r, -t, x^1, x^2)$, $i \to -i$, $(\phi, \bar{\phi}) \to (\bar{\phi}, \phi)$ and $A \to A$.[3] Similarly we could have defined an alternative parity transformations $P' = CP$.

The anti-linear $\mathcal{PT}$ symmetry is then indeed the product $P.T$ as defined above. Finally we note that the non-Hermitian boundary conditions eqs. (2.9) and (2.10) are $\mathcal{PT}$ invariant if $\xi$ is constant.

---

[3]This alternative definition was implicitly used in [32].

## B  Holographic renormalization.

In order to compute the 1-point functions of interest we need first to render the on-shell action finite. The renormalization is achieved via the inclusion of convenient covariant counterterms on the boundary which do not affect the dynamics of the system.

Let us first find the divergent contributions to the on-shell action. To do so we evaluate the bare action $S_0$ with the asymptotic expansion 2.20 and integrate in the radial direction up to some cutoff scale $\epsilon \ll 1$. For simplicity in this derivation we set $\lambda(v) = 0$. Thus we find

$$S_{reg} = \frac{1}{\kappa^2} \int_{u=\epsilon} d^3x \left[ -\frac{2}{\epsilon^3} - \frac{1}{2\epsilon}(1-\xi^2)\varphi_1^2 + \text{finite} \right]. \tag{B.1}$$

Now we ought to find a suitable covariant counterterm that kills those divergences. The standard way of proceeding is to invert the asymptotic expansion 2.20 and rewrite $S_{reg}$ in terms of the original fields. One arrives thus to

$$S_{ct} = \frac{1}{\kappa^2} \int_{\partial\mathcal{M}} d^3x \sqrt{-\gamma} \left( 2 + \frac{1}{2}\phi\overline{\phi} \right). \tag{B.2}$$

It is immediate to check that $S_{sub} = S_{reg} + S_{ct}$ is now finite on-shell. At this point we are able to follow the holographic prescription to obtain the 1-point functions:

$$2\kappa^2 \left\langle \mathcal{O}_\phi \right\rangle = 2\kappa^2 \lim_{\epsilon \to 0} \left( \frac{1}{\epsilon^{3-1}} \frac{1}{\sqrt{-\gamma}} \frac{\delta S_{sub}}{\delta\phi} \right) = \overline{\phi}_2 - \dot{\xi}\varphi_1 - (1+\xi)\dot{\varphi}_1 - iqa_0(1+\xi)\varphi_1, \quad \text{(B.3)}$$

$$2\kappa^2 \left\langle \mathcal{O}_{\overline{\phi}} \right\rangle = 2\kappa^2 \lim_{\epsilon \to 0} \left( \frac{1}{\epsilon^{3-1}} \frac{1}{\sqrt{-\gamma}} \frac{\delta S_{sub}}{\delta\overline{\phi}} \right) = \phi_2 + \dot{\xi}\varphi_1 - (1-\xi)\dot{\varphi}_1 + iqa_0(1-\xi)\varphi_1, \quad \text{(B.4)}$$

$$2\kappa^2 \left\langle J^v \right\rangle = 2\kappa^2 \lim_{\epsilon \to 0} \left( \frac{1}{\epsilon^3} \frac{1}{\sqrt{-\gamma}} \frac{\delta S_{sub}}{\delta A_v} \right) = -a_1 := -i\tilde{a}_1, \tag{B.5}$$

$$\kappa^2 \left\langle T_{vv} \right\rangle = \kappa^2 \lim_{\epsilon \to 0} \left( \frac{1}{\epsilon} \frac{2}{\sqrt{-\gamma}} \frac{\delta S_{sub}}{\delta\gamma^{vv}} \right) = -f_1 - \frac{1}{6}\varphi_1 \left( (1-\xi)\overline{\phi}_2 + (1+\xi)\phi_2 \right), \tag{B.6}$$

$$\kappa^2 \left\langle T_{x_1 x_1} \right\rangle = \kappa^2 \left\langle T_{x_2 x_2} \right\rangle = -\frac{1}{2}f_1 + \frac{1}{6}\varphi_1 \left( (1-\xi)\overline{\phi}_2 + (1+\xi)\phi_2 + 3\xi\dot{\xi}\varphi_1 \right). \tag{B.7}$$

## C  Numerical methods.

The equations of motion 2.12-2.19 have been solved numerically in the programming language *Julia* [36] by means of pseudospectral methods for the radial integration and a 4-th order Runge-Kutta mehtod for the time evolution. A useful introduction to pseudospectral methods may be found in [37]. The idea is to represent the radial dependence of all fields and their derivatives in a truncated basis of Chebyshev polynomials and solve for the expansion coefficients, reducing the problem to solving a sytem of equations, which in our case is linear. The radial gridpoints[4] are disposed in the so-called Chebyshev-Gauss-Lobatto grid, which is tailor designed to minimize the error in the solution. For sufficiently well-behaved functions the convergence grows exponentially with the number of gridpoints. Besides, these methods can deal with regular singular points, as it is the boundary $u = 0$. We have used $N_u = 20$ gridpoints in the radial direction. Checks have been made comparing to a bigger number of gridpoints with no significant difference observed. As for the time direction we used a timestep $\Delta v$ in the range $0.01 < 20N_u^2\Delta v < 1$ to avoid the so-called CFL numerical instabilities.

---

[4]Only an even number of gridpoints is allowed.

Another advantage is that one can simultaneously impose boundary conditions at the boundary and at the horizon. Convergence is further improved if one redefines the field subtracting explicitly the divergent terms which appear in the boundary expansion. It is also helpful that the fields vanish at most linearly as one approaches the boundary [28]. Thus we numerically work with

$$
\begin{aligned}
f(v,u) &:= \left(\frac{1}{u} + \lambda(v)\right)^2 + f_s(v,u),\\
g(v,u) &:= u^2 g_s(v,u),\\
a(v,u) &:= i a_s(v,u).
\end{aligned}
\tag{C.1}
$$

The arbitrariness in the radial shift function $\lambda(v)$ is exploited to keep the apparent horizon at a fixed location $u_h = 1$. As a result the domain of integration, which should include from the boundary to the horizon, becomes rectangular and boundary conditions at the horizon are easily implemented. In a metric given by 2.3 the apparent horizon is defined through the condition $dS(v,u_h) = 0$. Forcing it to lie at $u_h = 1$ gives the condition

$$
\dot{\lambda} + \frac{1}{2} e^{-g(v,1)} f(v,1) = 0.
\tag{C.2}
$$

In order to keep the horizon fixed at all times the previous condition should be time independent, more precisely $\partial_v dS(v,u)|_{u=1} = 0$. A straightforward computation allows to rewrite it as

$$
2f\left(6 - m^2 \phi\overline{\phi} - \frac{V}{2}\phi^2\overline{\phi}^2 - e^{2g} u^4 a'^2\right) + (d\phi - iqa\phi)(d\overline{\phi} + iqa\overline{\phi})\bigg|_{u=1} = 0,
\tag{C.3}
$$

which is understood as a boundary condition at the horizon for $f(v,u)$. Thus one is able to dynamically evolve $\lambda$ by computing $\dot{\lambda}$ either by direct computation of C.2 or by reading it off of the asymptotic expansion 2.20. Regarding the boundary conditions for the remaining fields, they can also be extracted from 2.20 in the standard manner.

Now let us discuss the numerical algorithm. The system of equations 2.12-2.19 contains three constraints, i.e. three equations with no time derivatives in it, and five dynamical equations. It can be shown that the time derivatives of the constraints are implied by the remaining equations, so that they are guaranteed to be satisfied so long as they are satisfied at some initial time. Alternatively, and we will do so, one can solve 2.12-2.16 at each time and regard the leftover equations as constraints which are satisfied at every point provided they hold at the boundary, for their radial derivative is implied by 2.12-2.16. Clearly this second alternative is easier to implement, not only the equations of motion are simpler but the system is decoupled if one solves for $(g, f, a, d\phi, d\overline{\phi})$. The time derivative of both scalar fields are extracted from $(d\phi, d\overline{\phi})$. It should be noted that even though the system of equations is uncoupled, the boundary condition C.3 couples $(f, d\phi, d\overline{\phi})$ non-linearly. In order to skip complications derived from it we simply estimate the values of $(d\phi, d\overline{\phi})$ at the horizon with an extrapolation formula.

The initial equilibrium state is prepared by introducing some seed non-equilibrium state[5] and evolving it until it relaxes down. Once the system is completely equilibrated we implement the non trivial profile of $\xi(v)$. The initial value of $\lambda(v)$ cannot be found *a priori* and a shooting method was introduced to that end.

Hence one solves the system of equations and obtains the sought subleading values from the profiles of the fields at each time.

---

[5] We set $\varphi_1 = 1$, $f_1 = -2.0$ and give an initial radial profile $\phi = (1 - \xi_i)\varphi_1 u$ and $\overline{\phi} = (1 + \xi_i)\varphi_1 u$.

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
