# Peer review of "Non-Hermitian Quantum Quenches in Holography"

_SciPost Physics, doi:SciPost Phys. 14, 030 (2023)_

## Round 1 · Referee Report · Anonymous (Referee 1) · 2022-5-14

Report

In [32] one of the authors used gauge/gravity duality to study PT symmetric quantum field theory. The current manuscript extends this work by considering time-dependent couplings. Interesting relations are found between non-unitary evolution in holographic quantum field theory and violation of the Null Energy Condition in the dual bulk theory, and various types of time-dependences are explored numerically. This is a new line of research, of which it is hard to predict where exactly it will lead, but I find it interesting.

I would like to request, however, that the authors explain more clearly the relation between PT symmetry in their holographic model and in previously studied PT symmetric quantum field theories. I find the paragraph starting at the bottom of page 10 (starting with “Let us also note how the PT symmetry acts in this case”) not clear enough. Hopefully the following specific questions will help convey what confuses me:

1) About the notation: Is there a reason why straight P and T are used, but the combination seems to be denoted as curly PT?
2) It is stated that the authors take the point of view that the PT symmetry only acts on the boundary conditions of the bulk theory, i.e., on the couplings of the dual field theory. Does this mean that PT does not act on the fields of the dual field theory (in contrast to what one might have expected of the parity and time reversal operators)?
3) For comparison, in arXiv:1201.1244, the iϕ3 theory is studied, where ϕ is assumed to be a pseudoscalar. It is stated that this interaction term is invariant under PT because P changes the sign of ϕ and T changes the sign of i. How is this action of P and T related to the point of view in the current manuscript?
4) In the manuscript, it is stated that the imaginary part of a complex scalar is a pseudoscalar and thus parity and time reversal odd. Could the authors explain why the imaginary part of a complex scalar is automatically a pseudoscalar? And could they clarify how the sign change of this pseudoscalar under T relates to the previous point 3) and to the discussion in [32], where T does not appear to change the sign of the imaginary part of the scalar? Perhaps relatedly, could the authors explain in more detail why the complexified boundary conditions are PT symmetric?
5) When describing the actions of P and T, the arguments of the scalar field are not displayed. I would have expected that T maps the boundary condition of a scalar field at a certain time to the boundary condition of the field at the reflected time. If the boundary conditions are allowed to display arbitrary time-dependence, how can they be PT symmetric? Is there a hidden assumption that boundary values at a certain time are the same as those at the reflected time?

  • validity: -
  • significance: -
  • originality: -
  • clarity: -
  • formatting: -
  • grammar: -

Author:  Sergio Morales-Tejera  on 2022-09-09  [id 2802]

(in reply to Report 1 on 2022-05-14)

We thank the referee for a careful reading of our manuscript and valuable suggestions. We have followed his advice and included an appendix in the new version of the manuscript with a more detailed explanation addressing the action of the discrete symmetries on the fields and comparing to previous conventions in the literature.

---

## Round 1 · Referee Report · Anonymous (Referee 2) · 2022-5-26

Report

This paper explores the idea of non-Hermitian holography. This is inspired by the observation that it is possible to have a non-Hermitian Hamiltonian with real eigenvalues, provided it has PT symmetry. Such theories carry the interpretation of an open quantum system, and in the PT-symmetric phase there is a steady balance between inflow and outflow.

The authors take a holographic superconductor model, and fix boundary conditions to engineer non-Hermitian field theory couplings. They then consider quench protocols involving the non-Hermitian coupling, solving the 1+1 bulk equations in ingoing coordinates.

I think this is a valuable contribution to a novel and interesting research direction (extending [32]) and this paper should be published in SciPost. But I think there is one aspect that should be clarified first, especially given the open quantum system interpretation:

Apparent and event horizons will eventually be the same once the system returns to equilibrium at late times, after the quench is over. Normally, if the null curvature condition is obeyed and you have an apparent horizon, then the event horizon has to be outside it or coinciding with it [see e.g. Hawking and Ellis]. Figure 3 shows the apparent horizon shrinking. Given the exotic bulk matter it is not at all clear whether the event horizon also shrinks, since we don't know for sure that it has to be outside the apparent horizon. So my question is - does the event horizon shrink? and, does it lie inside the apparent horizon for some time? Integrating null geodesics backwards through their computed solution to find the event horizon would answer this.

  • validity: -
  • significance: -
  • originality: -
  • clarity: -
  • formatting: -
  • grammar: -

Author:  Sergio Morales-Tejera  on 2022-09-09  [id 2801]

(in reply to Report 2 on 2022-05-26)

We thank the referee for the careful reading of our manuscript and for its valuable suggestions. We have followed its advice and addressed the question regarding the location of the event horizon, which shall be available in the discussion section of the new version of the manuscript.

---

## Round 2 · Referee Report · Anonymous (Referee 1) · 2022-9-21

Report

The authors have addressed my request. I recommend that the paper be published.

---

## Round 2 · Referee Report · Anonymous (Referee 2) · 2022-9-26

Report

In response to my query the authors have raised a potential issue about the validity of numerical approaches based on integrating just past the apparent horizon when NEC is violated. The authors have stressed that this requires a careful treatment. I think it is important that this has been highlighted for future analyses.

I recommend publication.

---

## Round 2 · List of Changes

Appendix on discrete symmetries added. Discussion regarding the event horizon added.

---

## Editorial Decision

published